# Deep Compression of Pre-trained Transformer Models

**Naigang Wang**      **Chi-Chun Liu**      **Swagath Venkataramani**      **Sanchari Sen**

**Chia-Yu Chen**      **Kaoutar El Maghraoui**      **Vijayalakshmi Srinivasan**

**Leland Chang**
IBM T. J. Watson Research Center
Yorktown Heights, NY 10598, USA
{nwang,cliu,swagath.venkataramani,sanchari.sen,
cchen,kelmaghr,viji,lelandc}@us.ibm.com

## Abstract

Pre-trained transformer models have achieved remarkable success in natural language processing (NLP) and have recently become competitive alternatives to Convolution Neural Networks (CNN) and Recurrent Neural Networks (RNN) in vision and speech tasks, respectively. Due to their excellent computational efficiency and scalability, transformer models can be trained on exceedingly large amounts of data at the expense of tremendous growth in model size. As high performance, large-scale, and pre-trained transformer models become increasingly available for users to download and fine-tune for customized downstream tasks, their deployment becomes challenging due to the vast amount of operations and large memory footprint. To address this challenge, we introduce methods to deeply compress pre-trained transformer models across three major application domains: NLP, speech, and vision. Specifically, we quantize transformer backbones down to 4-bit and further achieve 50% fine-grained structural sparsity on pre-trained BERT, Wav2vec2.0, and Vision Transformer (ViT) models to demonstrate 16x compression while maintaining model accuracy. This is achieved by identifying critical initialization strategies for quantization- and sparsity- aware fine-tuning as well as developing novel techniques such as quantizers with a zero-preserving format and scheduled dropout. These hardware-friendly techniques need only to be applied in the fine-tuning phase for downstream tasks, which renders them especially suitable for acceleration and deployment of pre-trained transformer models.

## 1   Introduction

Since its inception in 2017, the attention-based transformer architecture [1] has excelled in many Natural Language Processing (NLP) tasks and become the preferred building block for state-of-the-art language models. With their excellent capacity for transfer learning, transformer models that are pre-trained on large corpora, such as BERT [2]) and GPT-3 [3], can be fine-tuned for downstream tasks to achieve state-of-the-art performance. Today, fine-tuning a transformer-based pre-trained language model has become the de facto standard for NLP tasks. Inspired by this success in NLP, transformer models are also being actively explored in other application domains and making tremendous progress. For vision applications, transformer models have demonstrated excellent performance — matching or exceeding CNN-based models on image classification  [4, 5] and object detection [6] tasks. For speech applications, models with a transformer backbone such as Wav2vec2.0  [7], HuBERT [8], and

36th Conference on Neural Information Processing Systems (NeurIPS 2022).

WavLM [9] show comparable performance to traditional RNN/LSTM-based models, but with much improved computation efficiency and scalability.

Pre-trained transformer models used as a 'Swiss Army knife' for a wide range of machine learning tasks are becoming so-called "Foundation Models", which can potentially change the landscape of deep learning research and development [10]. Today, transformer models are pre-trained on increasingly large amounts of data, which has grown model sizes to unprecedented levels. In the past few years, many high performance, large-scale models with millions to billions parameters have been released for users to download and fine-tune for customized downstream tasks [11]. However, the vast number of computation operations and large memory footprint are becoming key barriers to their practical deployment and adoption in production settings. To address this challenge, techniques such as knowledge distillation [12–16] have been used to pre-train compact transformer-like models. However, standard transformer topologies remain the preferred architecture to achieve high performance on a wide range of tasks.

To compress deep neural networks, quantization has proven to be effective in many application domains and is a common technique to accelerate DNNs on GPU and other AI hardware platforms [17]. Thus far, quantization of transformer models has primarily been studied for NLP applications. 8-bit quantization was first applied to BERT in [18, 19] and showed negligible model accuracy degradation. Later, weight precision was pushed to 4-bit or lower [20, 21], but activations remained in 8-bit due to sensitivity to quantization errors. Recently, knowledge distillation has been shown to be effective in quantizing BERT models down to 4-bit for both weights and activations while maintaining baseline accuracy. Vision transformers (ViT) have been successfully quantized down to 6-bit [22] using post-training quantization (PTQ) techniques. For speech, the authors in [23] quantized a simplified transformer down to 8-bit. To date, vision and speech transformer models have not yet been successfully quantized down to precisions as low as 4-bit while NLP transformer models have only reached this level with complex distillation techniques.

Pruning is another promising method to compress transformer models — motivated by the assumption that the rich features learned during the pre-training phase may be redundant when fine-tuning for downstream tasks and can thus be pruned. In the past few years, pruning has been extensively explored on transformer models for NLP [24], ViT [25], and Speech [26] to achieve 40-50% sparsity without significant accuracy loss. However, most such work focuses on unstructured sparsity, which is challenging to harness in modern efficient hardware using vector- and matrix-based instructions. Hardware vendors are instead introducing techniques such as fine-grained structured sparsity [27], which can achieve hardware efficiency improvements across a wide range of DNN models.

Applying both quantization and pruning techniques to the same model is a particular challenge — due primarily to difficulties in coordinating two separate lossy mechanisms to optimize and recover model accuracy [28] [29]. To our knowledge, [27] is the only work to explore the combination of quantization and sparsity on pre-trained transformer models such as BERT. In those results, for 50% sparsity, INT8 is the lowest precision that achieves reasonable accuracy. To recover model accuracy in the sparse INT8 model, pre-training must be repeated with sparsity before fine-tuning for downstream tasks — a requirement that becomes extremely challenging due to access limitations to pre-training datasets and the significant compute resources needed for today's large-scale models [3].

In this paper, we combine 4-bit quantization and 50% pruning to deeply compress transformer models across multiple application domains, i.e. BERT, Wav2vec2.0, and Vision Transformer (ViT) models. Specifically, transformer backbones are quantized down to 4-bit and combined with 50% fine-grained structural sparsity to achieve up to 16x model compression while maintaining model accuracy. Our method does not require repeating pre-training nor accessing pre-training datasets. Instead, quantization and pruning are performed during the fine-tuning phase on downstream tasks using downloaded pre-trained models. Our techniques are straightforward to implement without the need for complex methods such as knowledge distillation, which may require large memory and compute resource. Our precision format and sparsity configuration are hardware-friendly and can be efficiently implemented in modern deep learning accelerators. Our methodology generalizes across different application domains, which brings an opportunity to greatly simplify the software and hardware-based acceleration and deployment of pre-trained transformer models.

In summary, we make the following contributions towards accurate, sparse, 4-bit transformer models across application domains:

1. We report, for the first time, transformer backbones quantized down to 4-bit precision with 50% fine-grained structural sparsity on pre-trained BERT, Wav2vec2.0, and ViT models. We achieve 1.62%, 0.45% and 0.52% degradation for the question-answering task on SQuAD1.1, the speech recognition task on Librispeech, and the image classification task on ImageNet1k, respectively. Without sparsity, all 4-bit models achieve $< 1\%$ accuracy loss with respect to the baseline.

2. We propose procedures for quantization-aware and sparsity-aware fine-tuning to minimize the impact of these two lossy processes. In particular, we show that proper initialization of the quantized model, quantization scales, and sparse model are critical to minimize accuracy losses.

3. We introduce novel techniques, including a SAWB+ weight quantizer with a symmetric, zero-preserving quantization format and scheduled dropout, to enable the deep compression.

## 2    Quantization-aware and Sparsity-Aware Fine-Tuning

Performing quantization/sparsity-aware training during the pre-training phase on large amounts of data may offer the best opportunity to optimize and recover model accuracy for extremely compressed transformer models [27]. However, in practice, the pre-training phase is not generally transparent to end users as it is often difficult to access pre-training data and repeat the pre-training phase, which may require hundreds of GPUs and very long training times [11]. On the other hand, the fine-tuning phase for downstream tasks is often lightweight, during which users can directly operate on their own datasets and tune models for only a few epochs [11]. In this work, we thus focus on quantization/sparsity-aware fine-tuning and introduce techniques in this context to enable deep compression of pre-trained transformer models.

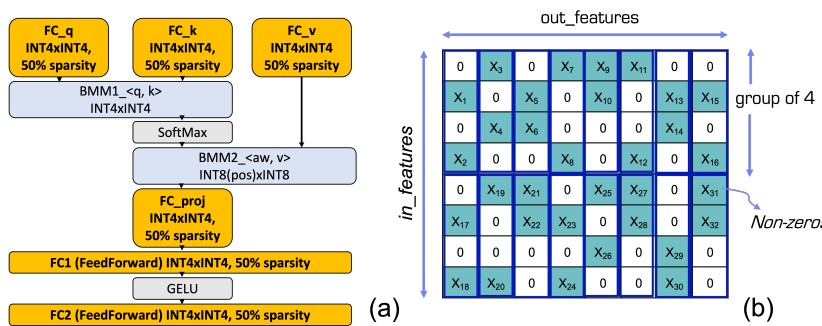

Figure 1: a) The transformer encoder block consisting of FC modules and BMM modules. Gray modules are linear layers where pruning is applied as shown in b), which is a schematic of a fine-grain structurally pruned weight matrix.

Figure 1 shows operations that are quantized and sparsified in a transformer block. Six linear layers, including QKV layers, a projection layer and two feed-forward layers are quantized to 4-bit and their weights are pruned by 50%. All input activations of two batch matrix-matrix product (BMM) layers are quantized. To preserve the fidelity of the probability distribution, we keep BMM2 in 8-bit.

### 2.1    Quantization

#### 2.1.1    Weight Quantization

**SAWB+** is an enhancement of the SAWB (statistics-aware weight binning) quantizer introduced by [30]. SAWB exploits both the first and second moments of the weight distribution to minimize the quantization error. The optimal quantization scale, $\alpha_w$ is calculated by the equation:

$$\alpha_w = c1 * \sqrt{E(w^2)} + c2 * E(|w|) \tag{1}$$

where c1 and c2 are coefficients empirically determined offline by fitting six standard distributions at a given precision. The details of extracting c1 and c2 can be found in [31] and the coefficients used in this paper are listed in Appendix-B. SAWB can effectively capture weight distributions to achieve excellent performance on a range of DNN models; however, in the backward path, SAWB

Table 1: The impact of zero-alignment on the performance of three sparse INT4 models.

| Zero alignment | BERT-base (F1 %) | Wav2vec2.0 (WER[1] %) | ViT (Accuracy%) |
|:---:|:---:|:---:|:---:|
| Yes | 87.07 | 4.53 | 83.60 |
| NO | 86.06 | 5.76 | 83.25 |

can cause instability. As shown in the left of equation 2, where $W$ and $W_q$ represent weight and quantized weight respectively, the clamped weights see zero gradients during optimization, which can cause two potential problems: 1) the variance of the weight gradients is reduced by truncation, which slows learning in non-quantized optimization, and 2) larger weights clamped early in training may never be updated, which reduces the representation power of the weight space during training. To overcome this problem, we introduce a simple modification to allow the gradients of clipped weights to pass through during weight updates while keeping the forward pass the same. This modified technique, SAWB+, captures the statistical characteristics of the distribution of weights just as in SAWB, but also has the ability to search the entire weight space during optimization with the same gradient variance per iteration step. In this work, we use SAWB+ for the weight quantization for all transformer models and observe no degradation from weight quantization.

$$\text{SAWB: } \partial W = \left\{ \begin{array}{ll} \partial Wq & |W| \leq \alpha \\ 0 & |W| > \alpha \end{array} \right. \quad \text{SAWB+: } \partial W = \left\{ \begin{array}{ll} \partial Wq & |W| \leq \alpha \\ \partial Wq & |W| > \alpha \end{array} \right. \tag{2}$$

**Zero alignment** is another critical technique to enable pruning at low precision. For integer weights, instead of utilizing the full range of precision levels, i.e. $2^k$, we choose to lose one level while keeping the distribution symmetric about zero, resulting in $2^k - 1$ levels. For example, in INT4, the number of integer levels is 15, i.e. [-7, -6, -5, -4, -3, -2, -1, 0, 1, 2, 3, 4, 5, 6, 7]. This way, floating point zeros in input will be represented by integer zeros. This zero alignment has significant impact on tuning the sparse model. As shown in Fig. 2, with zero-aligned weights, the BERT-base model converges to a much lower training loss and higher evaluation score in the fine tuning process as compared with the non-aligned case. The intuition is that with zeros in the weights as existing masks, the zero-aligned model requires less effort to learn the rest of the masks to achieve the same amount of sparsity. The same improvement is also observed in Wav2vec2.0 and ViT models, as shown in Table 1. Thus, just through zero alignment, the accuracy of three models can be boosted by 1.01%, 1.23% and 0.35%, respectively. Furthermore, symmetric and uniformly distributed quantization bins with zero alignment are also hardware friendly - allowing for simplified multiplication and accumulation (MAC) design and efficient data-flow [32].

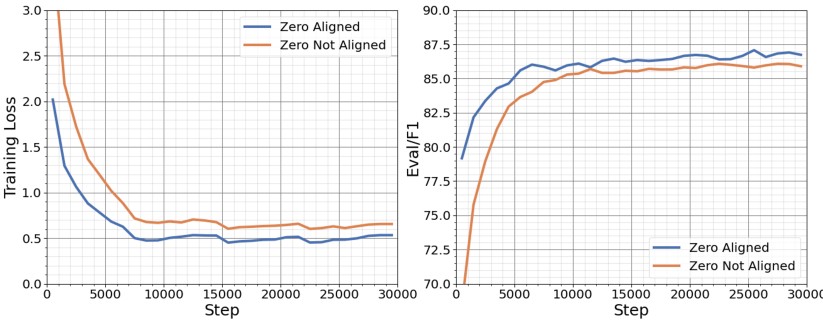

Figure 2: The impact of zero alignment on the convergence of sparse INT4 Bert-base on SQuAD1.1.

### 2.1.2 Activation Quantization

For activation quantization, we use two well-known quantizers: MinMax and PACT [31]. We extend the original 1-sided PACT to a 2-sided PACT to quantize activations with both positive and negative

---

[1]WER: Word error rate, lower the better.

values. $\alpha$ and $\alpha_n$ are used to define the dynamic range and the clamped activation output is uniformly quantized to k bits for the dot-product computation.

For the MinMax quantizer, $\alpha$ and $\alpha_n$ take the min and max of the tensors. For PACT, both clipping levels are parameterized and dynamically adjusted via gradient descent-based training. When using PACT, we sample 10 batches of the downstream dataset and initialize $\alpha$ and $\alpha_n$ of each layer based on a given percentile (e.g. 99% and 1%) of the observed activation distribution [30]. Note that we use the same quantizer for both symmetric and asymmetric activation functions such as GeLU. We also preserve activations of zero value by aligning zero to an integer level. Unlike for weights, the full $2^k$ levels are utilized for activation quantization.

While experimenting with different quantizers, we observed that for all INT8 models, the MinMax quantizer out-performed the PACT quantizer, as shown in Table 2. This observation comes as a surprise, since ideally, learned $\alpha$s should match the performance of the special MinMax case. As shown in Fig. 3, even when $\alpha$ is initialized to 99.9 or 99.95 percentile, training curves are noisier than when initializing to the max of the sampled activations. Further investigation led us to conclude that a small amount of large-amplitude outliers play an important role in transformer training compared to CNN-based models [33].

INT4 models are very limited in the number of precision levels available and hence become more sensitive to quantization errors as compared to INT8. For this reason, we use the MinMax quantizer for INT8 and sparse INT8 models while using PACT for INT4 models. The details of $\alpha$ initialization conditions are listed in Appendix-B.

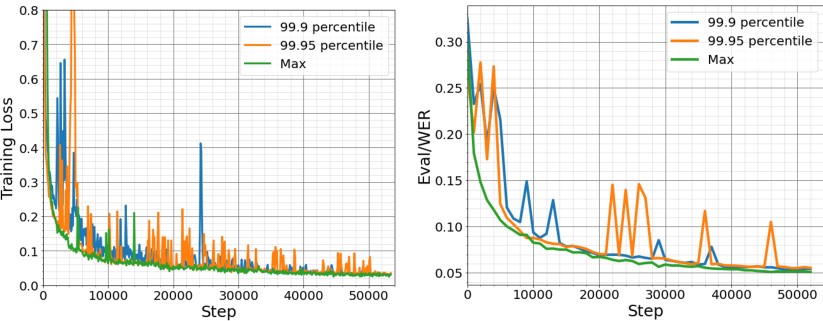

Figure 3: The impact of clipping $\alpha$ initialization on training loss and WER of sparse INT4 Wav2vec2.0 on Librispeech

## 2.2 Fine-Grained Structured Pruning

Figure 1b depicts the details of the fine-grained structured pruning performed in this work, which is based on the 2:4 sparsity proposed in [27]. We form groups of 4 consecutive weights values along the "in_features" dimension of weights and zero out 2 values to achieve 50% sparsity. The location of the two zero values can vary across the different groups in a layer. The pruned weights can be compressed by storing only the non-zero weight values and their corresponding indices within each group — 2 non-zero values and 2 indices per group. Since indices can only take one of four possible values for a group size of 4, 2 bits are sufficient to represent each index within a group.

We use a mask for weight pruning. The weight pruning mask is extracted only once before the start of the training instead of, e.g. Iterative Magnitude Pruning (IMP) that can be otherwise adopted [28] [29]. At each iteration, the weights are quantized first and the pruning mask is applied on the quantized weights and then fed into matrix computation.

## 2.3 Quantization- and Sparsity-Aware Fine-Tuning

A key challenge in combining quantization and sparsity is to determine the workflow by which to apply these two lossy processes during fine-tuning, e.g. tuning with quantization followed by pruning, or vice versa. We discover that quantization and sparsity can be jointly fine-tuned in one

Table 2: MinMax vs. PACT quantizers for INT8 models.

| Quantizer | BERT-base (F1 %) | Wav2vec2.0 (WER %) | ViT (Accuracy%) |
|-----------|------------------|---------------------|------------------|
| MinMax | 88.35 | 3.85 | 84.47 |
| PACT | 87.89 | 4.05 | 83.98 |

optimization process provided that the process is properly initialized. In this section, we explain the key initialization methods for the compressed model to achieve high accuracy.

### 2.3.1 Initialization for Quantization

Pre-trained models are generally trained on large datasets in full FP32 precision. The rich features learned during pre-training should provide opportunity for models to accommodate quantization error during the fine-tuning phase. However, when we use the pre-trained model as initialization for quantization-aware fine-tuning, the resulting quantized models consistently show inferior performance as shown in Table 3. For INT4 even without sparsity, the three models studied suffer significant degradation of 2.01%, 0.82%, 1.37%. Even in INT8, the ViT model incurs >1% degradation.

Instead, we can use full precision fine-tuned models to initialize quantization-aware fine-tuning. As shown in Table 3, this significantly boosts the performance of the quantized model both in INT8 and INT4 precisions. Particularly for INT4, all three models achieve accuracy within $< 1\%$ of the baseline. To the best of our knowledge, this is the first time that INT4 transformer models (both weight and activation) achieve iso-accuracy without the teacher-student distillation technique, which requires computation of the full precision teacher models during tuning rather than just using them for initialization. The results also indicate that initialization with pre-trained models is more vulnerable to quantization noise despite redundant features, whereas initialization with the full precision fine-tuned models effectively learn the features most relevant to downstream tasks, which improves the resultant quantized models. When combined with the quantizers described in Section 2.1, quantized model accuracy is significantly improved.

### 2.3.2 Initialization for Pruning

When adding sparsity, for INT8 precision models, the full precision fine-tuned models are effective as initialization, as shown in table 3. Both the Wav2vec2.0 and ViT models can reach baseline accuracy while the Bert-base model is within 1% of the baseline. Note that we achieve the same accuracy on a sparse INT8 BERT model as reported in [27], but without the need to repeat pre-training.

For sparse INT4 models, however, it becomes difficult to maintain accuracy with this same initialization. As shown in Table 3, BERT and ViT models, in particular, suffer significant degradation $> 2\%$. To overcome this, we propose a different initialization strategy, in which we use a pre-tuned sparse INT8 model as initialization to fine-tune the sparse INT4 model. This is inspired by the observation that INT8 models show no degradation when using MinMax quantizers. Intuitively, a pre-tuned sparse INT8 model is a sparse architecture that can already tolerate low precision, which makes it a strong starting point to further adjust to the larger quantization errors associated with INT4. As shown in Table 3, this initialization method boosts the accuracy significantly and brings Wav2vec2.0 and ViT sparse INT4 models to within 1% of baseline accuracy. Fig. 4 shows the convergence curves for the fine-tuning of sparse INT4 ViT models initialized using FP32 and sparse INT8 fine-tuned models. The training loss starts at a much lower level when using sparse INT8 initialization and the network converges to lower loss and higher accuracy across the entire tuning process.

### 2.3.3 Scheduled Dropout

For the BERT model, sparse INT8 initialization starts at lower training loss just as in Wav2vec2.0 and ViT, but it does not show immediate improvement in the F1 score due to over-fitting potentially caused by the small size of the SQuAD1.1 dataset. To effectively take advantage of sparse INT8 initialization and avoid over-fitting in the early stages of training, we propose scheduled dropout. Scheduled dropout initially applies a very high dropout rate, e.g. 0.35, and then linearly decreases the rate until the 10k-th iteration, where it finally stays constant, e.g. at 0.2, until the end of the fine-tuning

Table 3: The impact of initialization model on the accuracy of INT8 and INT4 models. FP32 means fine-tuned FP32.

| Precision | Initialization | BERT-base (F1 %) | Wav2vec2.0 (WER %) | ViT (Accuracy %) |
|---|---|---|---|---|
| INT8 | Pre-trained | 87.95 | 3.79 | 83.05 |
| | FP32 | 88.35 | 3.85 | 84.47 |
| INT4 | Pre-trained | 86.68 | 5.02 | 82.75 |
| | FP32 | 87.86 | 4.56 | 83.49 |
| INT8 + 50% sparsity | FP32 | 87.70 | 4.21 | 84.03 |
| INT4 + 50% sparsity | FP32 | 86.75 | 5.09 | 82.66 |
| | Sparse INT8 | 87.07 | 4.65 | 83.60 |

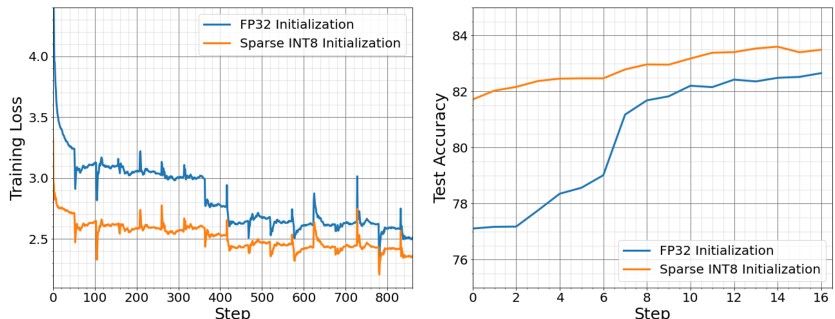

Figure 4: Convergence of sparse INT4 ViT on ImageNet1k using sparse INT8 vs. FP32 initialization.

phase. As shown in Fig. 5, scheduled dropout improves generalization from the beginning of the training, which enables 4-bit models to get closer to baseline F1 scores.

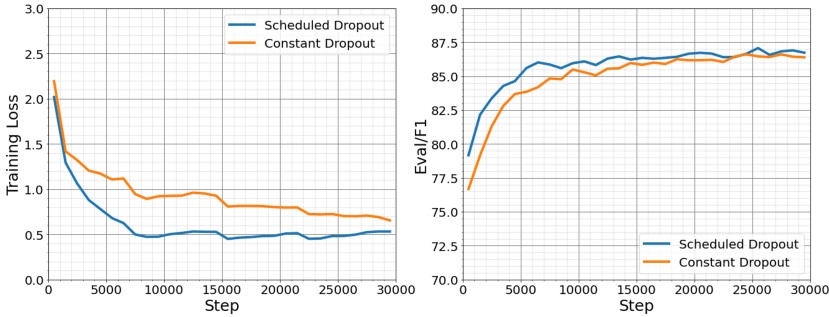

Figure 5: Scheduled vs. constant dropout on the convergence of sparse INT4 BERT on SQuAD1.1

## 2.4 Hardware Implementation

To evaluate the hardware benefits of quantized and pruned transformers, we consider a systolic-array-based AI hardware accelerator similar to [34] [35]. We enhance the accelerator to further exploit fine-grained structured weight sparsity by skipping computations on zero-valued weights. Specifically, the systolic array is enhanced to read only the non-zero weights in a group and perform a multiply-accumulate operation with the activation value corresponding to the non-zero index.

The primary hardware benefit of sparse transformers is the memory capacity and bandwidth savings obtained from storing their weights in a compressed manner (Section 2.2). These savings are a direct function of the amount of weight sparsity and the precision of weights. Accordingly, for INT8

precision with 50% sparsity, we achieve a memory capacity savings of $1.6\times$ with respect to a dense INT8 baseline and for INT4 precision with 50% sparsity, we achieve a memory capacity savings of $1.33\times$ with respect to a dense INT4 baseline, across all benchmarks. Overall, when compared to a dense FP32 baseline, a sparse INT8 transformer and a sparse INT4 transformer achieves $6.4\times$ and $10.67\times$ savings in memory capacity, respectively. In addition to the memory capacity savings, we also get performance benefits from only performing computations on the non-zero weight values. The overall performance benefits observed for a benchmark depends on the size of different layers in the transformer block and their execution time fraction. By pruning the 6 layers inside each transformer block shown in Figure 1, we affect their execution times while keeping the execution time for all other layers unchanged. We observe that the 6 pruned layers account for 57.7%, 52.4% and 51.1% of the overall execution time for BERT-base, Wav2vec2.0, and ViT respectively in their dense INT8 execution. In dense INT4 execution, these 6 layers account for 40.5%, 35.5% and 34.3% of the overall execution time for BERT-base, Wav2vec2.0 and ViT respectively. Accordingly, we achieve a performance improvement of $1.4\times$, $1.35\times$ and $1.34\times$ for BERT-base, Wav2vec2.0 and ViT respectively in INT8 with respect to dense models and a performance improvement $1.25\times$, $1.22\times$ and $1.21\times$ in INT4. When compared to a dense FP16 baseline, the performance improvements of sparse INT8 and INT4 further increase by a factor of $2.76\times$ and $3.78\times$ on an average across all benchmarks.

## 3 Experimental Setup and Results

We evaluate the proposed methods on three representative pre-trained models and corresponding downstream benchmarks in multiple application domains to demonstrate the effectiveness of our methods. Specifically, we investigate the BERT-base model on the SQuAD1.1 benchmark, the Wav2vec2.0 model on the Librispeech dataset and the ViT-base model on the ImageNet1k benchmark. We implement quantization and pruning on top of Huggingface Transformer [36] for BERT and Wav2vec2.0 models, and Timm packages [37] for the ViT model. We use the recipe published in the original papers to fine-tune the downstream benchmarks and use the results as the baseline to evaluate our method. As shown in Table 4, our baseline matches the published results in the corresponding paper or git repository. All experiments are performed on NVIDIA V100 GPUs. Listed below is basic information on the models and benchmarks — more details are presented in Appendix-A.

**Pre-trained BERT-base model on the SQuAD1.1**: The BERT-base model has 12 transformer blocks with 12 self-attention heads and a hidden dimension of 768. We use the pre-trained BERT model provided by Huggingface Transformer for fine-tuning SQuAD 1.1, a standard question-answering task for NLP. For fine-tuning, we use batch size of 12 and sequence length of 384. We use the AdamW optimizer with a learning rate of 3e-5 with linear decay. The model is fine-tuned for 2-4 epochs with a dropout probability of 0.1-0.2. For the sparse INT4 model, scheduled dropout is applied as introduced in section 2.3.3.

**Wav2vec2.0 large model on the Librispeech**: The Wav2vec2.0-large model contains 24 transformer blocks with 16 attention heads and a hidden dimension of 1024 [7]. We use the facebook/wav2vec2-large-lv60 pretrained model on HuggingFace Transformer, which is pre-trained on the audio data from LibriVox. The pretrained model is fine-tuned on Librispeech's 100 hour clean subset using standard Connectionist Temporal Classification (CTC) loss for the Automatic Speech Recognition (ASR) downstream task. The model has a varied token length averaging 600-800. For fine tuning, we use the AdamW optimizer with a learning rate of 3e-4. The learning rate decays linearly after 500 warm-up steps. We use a batch size of 32 and tune the model for 6 epochs.

**ViT-base model on ImageNet1k**: The ViT-base model contains 12 transformer blocks with 12 attention heads and a hidden dimension of 768 [4]. To clearly understand the impact of quantization/pruning on transfer learning, we use the pre-trained model that is only trained on ImageNet21k and then fine-tune it on ImageNet1k for downstream image classification. For fine-tuning, we use a patch size of 16x16 and fine-tune resolution of 384x384, following the original settings of [4]. The optimizer is SGD with a learning rate of 0.01. We tune the model for 8 epochs using a cosine learning rate schedule, gradient clipping of 1.0, and batch size of 512. For sparse INT8/4 models, we find that it is beneficial to stimulate training by increasing the learning rate to 0.05 and tune for 16 epochs until convergence settles.

Quantization and pruning are applied only on transformer backbones as they contribute $> 99\%$ of the total compute operations. We use a common setting for all three models and benchmarks.

Table 4: Results for deeply compressed BERT-base, Wav2vec2.0, and ViT models. FP32 refers to FP32 fine-tuned on downstream tasks. Sp is short for sparsity.

| Precision Sparsity | Weight Quantizer | Activation Quantizer | Initialization Model | BERT-base (F1%) | Wav2vec2.0 (WER %) | ViT (Accuracy %) |
|---|---|---|---|---|---|---|
| FP32 | – | – | Pre-trained | 88.69 | 4.20 | 84.12 |
| INT8 | SAWB+ | MinMax | FP32 | 88.35 (-0.34) | 3.85 (+0.35) | 82.47 (+0.35) |
| INT8+50%Sp | SAWB+ | MinMax | FP32 | 87.70 (-0.99) | 4.21 (-0.01) | 84.03 (-0.09) |
| INT4 | SAWB+ | PACT | FP32 | 87.86 (-0.83) | 4.53 (-0.33) | 83.49 (-0.63) |
| INT4+50%Sp | SAWB+ | PACT | INT8+50%Sp | 87.07 (-1.62) | 4.65 (-0.45) | 83.60 (-0.52) |

Specifically, the SAWB+ quantizer is used for weight quantization. The MinMax quantizer is used for the activation quantization for INT8 and sparse INT8 models, and the PACT quantizer is used for INT4 and sparse INT4 models. Full precision fine-tuned models are used for initialization of INT8, sparse INT8 and INT4 models. For the sparse INT4 model, we use a sparse INT8 model for initialization as discussed in section 2.3.2. More details of the training methodology are presented in Appendix-B. Table 4 shows the final results on the three benchmarks with different precision/sparsity settings. The uncertainty analysis of the results is presented in Appendix-B, where table 7 As shown, aside from the sparse INT4 BERT-base model, which shows a small degradation of $1.6\%$, all other compressed models achieve iso-accuracy within $< 1\%$ of their respective baselines, making them good candidates for efficient deployment.

# 4 Related Work

Quantization has been extensively studied to compress CNN and RNN models. Over the years, many approaches, such as quantization-aware training [31], post-training-quantizaiton [22], knowledge distillation [38], and quantizers, such as PACT [31], LSQ [39], LSQ+ [40], and SAWB [30], have been developed to improve the performance of quantized models. For NLP, these techniques have been applied to compress transformer models. FullyQT [18] and [41] developed an 8-bit transformer for machine translation. Q8BERT [19] and IBERT [42] quantize BERT and RoBERTa [43] models respectively in 8-bit. Later studies focus on lower precision weight quantization. QBERT [20] quantizes weights down to 4-bit without significant degradation using Hession information. TernaryBERT, BinaryBERT and BiBERT [44–46] pushed weight precision to even lower than 4-bit, while maintaining activations in 8-bit. Recently, KDLSQ-BERT [47] and MKQ-BERT [48] discovered that distillation techniques can be used to effectively quantize activations in the BERT model into 4-bit. Our methods, however, are more straightforward, easier to implement, and can achieve good performance while avoiding some of the limitations of distillation such as the requirement of storing and computing the teacher models. More importantly, our methods and distillation are mutual exclusive and can be combined to further improve the performance of compressed models.

There has been less work in quantizing vision and speech transformers, however, Bie et. al [23] quantize a transformer-based encoder-decoder architecture to 8-bit for the end-to-end ASR task, but the transformer is simplified with only 6 encoder/decoder layers and quantization is only applied to weights. Liu et al. [22] and FQ-ViT [22] quantize vision transformer models such as ViT [49], DeiT [38] and DETR [50] and achieve iso-accuracy at 8-bit precision, but noticeable degradation is observed when the precision is reduced down to 6-bit.

Pruning has been extensively applied to compress transformer models in NLP [51–53], ViT [54–56], and Speech [26]. However, most pruning approaches are unstructured, which can be challenging for hardware-based acceleration. Structured pruning in the form of directly removing attention heads [57, 58] shows promising results, but these methods may not be applicable to transformers in other application domains [55]. To the best of our knowledge, there is no common structural pruning approach that can be broadly applied.

The most relevant work is [27], which explores the combination of quantization and sparsity on transformer models and is the first to propose fine-grained structural pruning. However, the work is only verified on pre-trained BERT models, the lowest precision reached is 8-bit, and the sparse model must repeat pre-training to recover model accuracy. Using our approach, we achieve the same

accuracy while only impacting the fine-tuning phase. We also successfully push precision down to 4-bit and demonstrate our approach across three representative application domains.

# 5 Conclusion

Pre-trained transformer models have demonstrated state-of-the-art performance in a wide range of applications, but tremendous growth in computational requirements and memory footprint render deployment in practical applications a significant challenge. To address this, we introduce a novel quantization/sparsity-aware fine-tuning methodology to successfully compress pre-trained transformer models by carefully choosing suitable quantizers, data format, initialization, and regularization techniques. Our methods are demonstrated on popular pre-trained transformer models and benchmarks across three application domains. This work is critical as it lays the foundation to deploy large pre-trained transformer models in practical end-user applications including data centers and edge scenarios. Our inference solutions can accelerate ML deployment and provide significant cost and energy savings for AI inference. Our solutions could still be subject to unexpected instabilities, and going forward, will require task-specific robustness studies to prepare these models against adversarial attacks. More details on the broader impact of our work are addressed in Appendix-C.

## Acknowledgments and Disclosure of Funding

This work is fully funded by IBM Research. The authors would like to thank I-Hsin Chung, Ming-Hung Chen, Paul Crumley and James Norris for their support on computing infrastructure; Xiao Sun, Kailash Gopalakrishnan, Yuhong Li, Xiaodong Cui, Andrea Fasoli, Jiamin Ni, Derrick Liu and Mauricio Serrano for the technical discussions; Jeffrey Burns and Mukesh Khare for their executive support and feedback on the draft; IBM AI Hardware Center for providing computing resources.

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
