# A   Appendix-A: Model and Dataset

In this section, we provide details on the datasets, model architectures, and settings used for full precision fine-tuning to replicate the baseline accuracy reported in prior publications.

## A.1   BERT-base/SQuAD1.1

**SQuAD1.1** is the Stanford Question Answering dataset [1]. It is a reading comprehension dataset, consisting of a collection of 100k question and answer pairs from reading passages, where the answer to every question is a segment of corresponding passage. The task is to predict the answer text span in a passage.

**BERT-base** model [2] is a transformer model pre-trained on a large corpus of English data, BooksCorpus [3], with 800M words and English Wikipedia of 2500M words. The model was pre-trained in a self-supervised fashion using a masked language modeling (MLM) procedure. The BERT-base model consists of an input embedding, 12 transformer blocks, and an output linear layer, with a total of 100M parameters. The input embedding is a sum of token embeddings, segmentation embeddings and the position embeddings. Each transformer block contains 12 self-attention heads and a hidden size of 768.

To fine-tune BERT-base on the SQuAD1.1 downstream task, we use a batch size of 12 and a sequence length of 384. The experiments are performed on 4 V100 GPUs with a per-gpu batch size of 3. For the full precision fine-tuning baseline, we follow the fine-tuning strategy from [2] and use the AdamW optimizer with a learning rate of 3e-5 with linear decay. The model is fine-tuned for 2 epochs with a dropout probability of 0.1. We obtain a baseline F1 score of 88.69, which closely matches the F1 score (88.50) published in [2]. Fig. 1a shows the convergence curve of the full precision fine-tuning baseline.

## A.2   Wav2vec2.0/Librispeech

**Librispeech** is a corpus of English speech for the automatic speech recognition (ASR) task [4]. It contains 1000 hours of speech sampled at 16 kHz. The training data is split into 3 partitions of 100hr, 360hr and 500hr sets with 'clean' and 'other' categories. In this work, we use the 100hr-clean data for the downstream task and the clean validation subset for evaluation.

**Wav2vec2.0-large** is a speech model pre-trained on the audio data from LibriVox (LV-60k) [5] in a self-supervised manner [6]. In this work, we use the Wav2vec2.0-large model, which has a large transformer backbone with 24 transformer blocks. The hidden dimension, inner dimension, and number of attention heads in each transformer block are 1024, 4096 and 16, respectively. Prior to being fed into the transformer backbone, the raw waveform is encoded through multiple 1d-convolution layers followed by layer normalization and GeLU activations.

The pre-trained model is fine-tuned on Librispeech's 100 hour clean subset using standard Connectionist Temporal Classification (CTC) loss. We follow the implementation and settings from HuggingFace Transformer [7] for the fine-tuning. Specifically, for the full precision fine-tuning baseline, we use the AdamW optimizer with betas=(0.9,0.999) and a learning rate of 3e-4. The learning rate decays linearly after 500 warm-up steps. We use 8 V100 GPUs to tune the model for 3 epochs with a total batch size of 32. We achieve a baseline Word Error Rate (WER) of 4.20 %, matching the result provided by HuggingFace Transformer (4.2 %). Fig. 1b shows the convergence curve of the full precision fine-tuning baseline.

## A.3   ViT/ImageNet1k

**ImageNet1k** [8] is an image classification benchmark which consists of 1000-categories of objects with over 1.2M training and 50K validation images.

**ViT-base** model is a BERT-like transformer encoder model taking images as the input for image classification tasks. The input images are split into fixed-sized patches of 16x16 and linearly embedded. The ViT-base model has 12 transformer blocks with 12 attention heads and a hidden dimension of 768 [9]. In this paper, we use the pre-trained model that is pre-trained on ImageNet21k [8] and then fine-tune it on ImageNet1k for downstream image classification. We use a resolution of 384x384 for

Table 1: Coefficients for SAWB+.

| Precision | C1 | C2 |
|-----------|--------|--------|
| INT4 | 12.68, | -12.80 |
| INT8 | 31.76, | -35.04 |

fine-tuning following the original settings of [9]. The optimizer is SGD with a learning rate of 0.01. We tune the model for 8 epochs using a Cosine learning rate schedule, gradient clipping of 1.0, and batch size of 512 on 32 V100 GPUs. With these settings, we achieve an accuracy of 84.12, which matches the accuracy (83.97) published in [9]. Fig. 1c shows the convergence curve of full precision fine-tuning baseline.

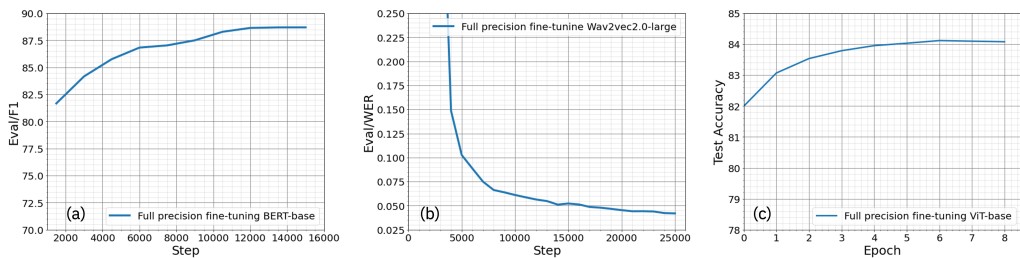

Figure 1: Convergence curves of full precision fine-tuning of (a) BERT-base on SQuAD1.1; (b) Wav2vec2.0-large on Librispeech; and (c) ViT-base on ImageNet1k.

## B    Appendix-B: Quantization- and Sparsity-aware Fine-tuning Settings

In this section, we provide details on the implementation of quantization and pruning operations, as well as the hyper-parameters used for the fine-tuning of deeply compressed models.

### B.1    Quantization/Pruning Implementation

We implement quantization and pruning in the PyTorch framework. For each linear module or batch-matrix-matrix multiplication (bmm) operation, we insert quantization operations to quantize both the activations and weights. Fig. 2 shows screenshot examples of quantized modules with inserted quantization operations from the graph of the quantized BERT-base model. For linear modules, both input activations and weights are quantized, as shown in Fig. 2(a) a query layer in self-attention and Fig. 2(c) an intermediate-dense layer in the feed-forward network (FFN); for bmm operations, both input activations are quantized as shown in Fig. 2(b).

Fig. 3 presents a toy example showing a deeply compressed linear module, i.e. QLinear, running a forward pass with random input. The linear layer is quantized in 4-bit for both weights and activations using SAWB+ and PACT quantizers, respectively. Weights are further pruned with 50% sparsity using a fine-grained group of 4 as discussed in section 2.2. The printout shows the pruning mask tensor, pruned weight tensor, and quantized weight tensor computed during the forward pass.

### B.2    Fine-tunine Setting

We use a common setting for all three models and benchmarks. Full precision fine-tuned models are used for the initialization of INT8, sparse INT8, and INT4 models. For the sparse INT4 model, we use a sparse INT8 model for initialization as explained in section 2.3. Fig. 4 shows a schematic of the fine-tuning procedures. The SAWB+ quantizer is used for weight quantization for all models as discussed in section 2.1.1. The coefficients used in SAWB+ are listed in Table  1. The MinMax or PACT quantizer is used for the activation quantization. For the PACT quantizer (discussed in section 2.1.2), three hyper-parameters are used to train $\alpha$ and $\alpha_n$ parameters, i.e. initiation in percentile,

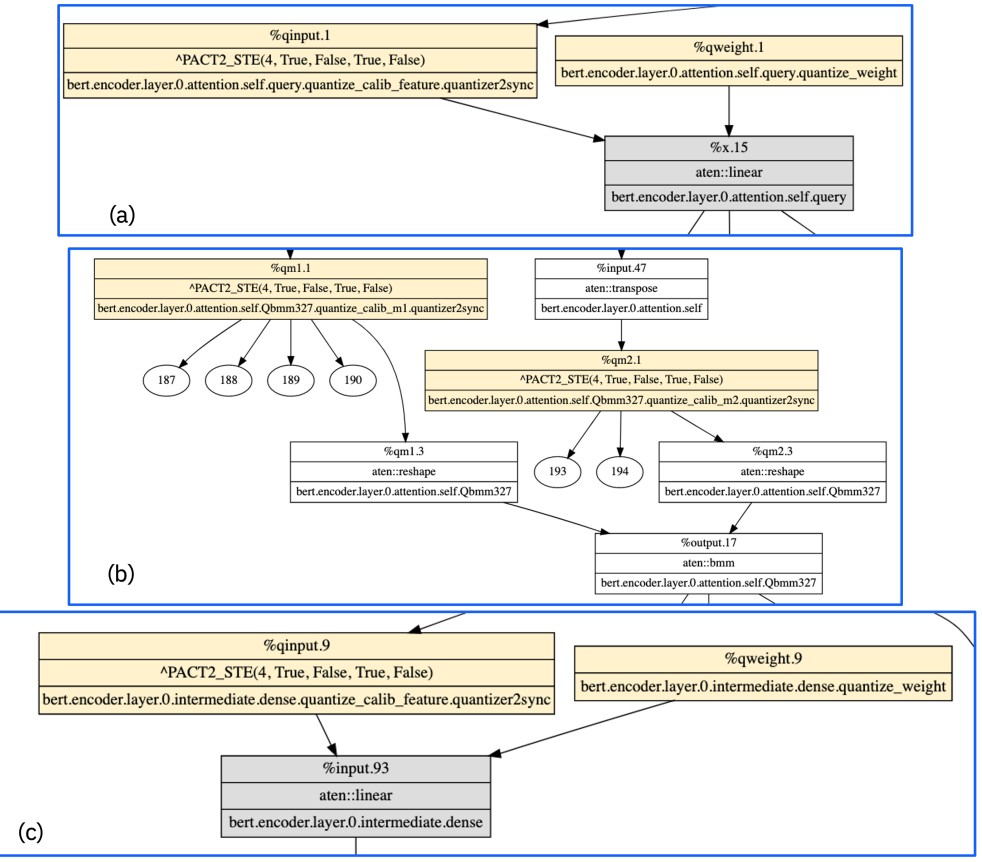

Figure 2: a) Screenshot examples of a quantized graph with implemented weight and activation quantization operations, for (a) a query linear layer; (b) a bmm operation for attention computation; and (c) an intermediate dense linear layer in FFN from layer0 (the first transformer block) of the deeply compressed BERT-base model.

Table 2: Qantization/Sparsity-aware fine-tuning setting for BERT-base on SQuAD1.1. "Sp" indicates sparsity.

| Precision Sparsity | Weight Quantizer | Activation Quantizer | Initialization Model | Percentile (%) | $\alpha$_lr | $\alpha$_decay | Dropout |
|---|---|---|---|---|---|---|---|
| INT8 | SAWB+ | MinMax | FP32 | – | – | – | 0.2 |
| INT8+50%Sp | SAWB+ | MinMax | FP32 | – | – | – | 0.2 |
| INT4 | SAWB+ | PACT | FP32 | 99 | 1e-3 | 1e-3 | 0.2. |
| INT4+50%Sp | SAWB+ | PACT | INT8+50%Sp | 99 | 1e-3 | 1e-3 | Scheduled |

learning rate ($\alpha$_lr) and L2 decay ($\alpha$_decay). The detailed settings used for three benchmarks are as follows.

Table 2 lists the settings for BERT-base/SQuAD1.1 benchmark. We use the same baseline optimization methods as described in Appendix 1.1, except that the compressed models are fine-tuned for 4 epochs with a larger dropout (0.2) or a scheduled dropout as introduced in section 2.3.3. Fig. 5 shows the convergence curves of the deep compressed models.

Table 3 lists the settings for Wav2vec2.0-large/Librispeech benchmark. We use the same baseline optimization methods as described in Appendix 1.2, except that the compressed models are fine-tuned for 6 epochs. Fig. 6 shows the convergence curves of the deeply compressed models.

Table 4 lists the settings for ViT-base/ImageNet1k benchmark. For INT8/4 models without pruning, we use the same baseline optimization methods as described in Appendix 1.3. For sparse INT8/4

```
1  #a toy example to quantize and prune one Linear layer with dummy inputs
2  model_q = QLinear(in_features=64, out_features=4, num_bits_feature=4, num_bits_weight=4, \
3                    p_group=4, p_ratio = 0.5, qa_mode='pact', qw_mode='sawb+')
4  #get pruning mask
5  model_q.get_mask()
6  output = model_q(torch.rand(1, 4, 64))
```

```
Weights[:,:8]:
 tensor([[ 1.1034e-01,  7.0334e-02, -9.7582e-02,  4.9689e-02, -8.1956e-07,
          -9.7793e-02, -8.3745e-06, -1.0821e-01],
         [-1.1906e-01,  8.3641e-02,  3.2591e-02,  3.1367e-02, -1.5860e-02,
           6.3644e-02, -1.1322e-01, -1.1617e-01],
         [ 2.6805e-02, -3.2602e-02,  1.0721e-01, -3.6630e-02, -1.1574e-01,
           7.5815e-02, -5.7634e-02, -1.1444e-02],
         [ 1.1820e-01,  5.0626e-02, -7.3768e-02,  6.0336e-02, -8.2874e-02,
          -6.0145e-02,  1.0179e-01,  9.6717e-02]])
Mask[:,:8]:
 tensor([[1., 0., 1., 0., 0., 1., 0., 1.],
         [1., 1., 0., 0., 0., 0., 1., 1.],
         [0., 0., 1., 1., 1., 1., 0., 0.],
         [1., 0., 1., 0., 0., 0., 1., 1.]])
Pruned weight[:,:8]:
 tensor([[ 0.1103,  0.0000, -0.0976,  0.0000, -0.0000, -0.0978, -0.0000, -0.1082],
         [-0.1191,  0.0836,  0.0000,  0.0000, -0.0000,  0.0000, -0.1132, -0.1162],
         [ 0.0000, -0.0000,  0.1072, -0.0366, -0.1157,  0.0758, -0.0000, -0.0000],
         [ 0.1182,  0.0000, -0.0738,  0.0000, -0.0000, -0.0000,  0.1018,  0.0967]])
Quantized weight[:,:8]:
 tensor([[ 0.1067,  0.0000, -0.1067,  0.0000,  0.0000, -0.1067,  0.0000, -0.1067],
         [-0.1067,  0.0711,  0.0000,  0.0000,  0.0000,  0.0000, -0.1067, -0.1067],
         [ 0.0000,  0.0000,  0.1067, -0.0356, -0.1067,  0.0711,  0.0000,  0.0000],
         [ 0.1067,  0.0000, -0.0711,  0.0000,  0.0000,  0.0000,  0.1067,  0.1067]])
```

Figure 3: a) A toy example of a quantized and pruned linear module running a forward pass with a random input. The printout shows the pruning mask, pruned weight and quantized weight tensors.

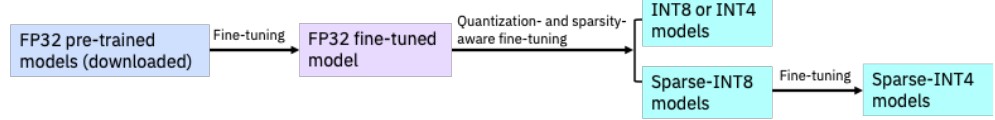

Figure 4: A schematic of the fine-tuning procedures leading to deep compressed models. Quantization- and sparsity- aware fine-tuning are initialized by FP32 fine-tuned models to obtain the INT8, INT4 or sparse-INT8 models. The sparse-INT8 models are further fine-tuned to achieve sparse-INT4 models.

models, we tune the model for 16 epochs with an initial learning rate of 0.05, keeping the rest of hyper-parameters the same as the baseline. Fig. 7 shows the convergence curves of the deep compressed models.

### B.3   Error Bar

To understand the uncertainty of the results, we run five experiments with random initial seeds for all three sparse INT4 models. The means and standard deviations calculated using the five runs are shown in the Table 5. The results are consistent with Table **??**.

Table 3: Qantization/Sparsity-aware fine-tuning setting for Wav2vec2.0-large on Librispeech. Sp is short for sparsity.

| Precision Sparsity | Weight Quantizer | Activation Quantizer | Initialization Model | Percentile (%) | $\alpha$_lr | $\alpha$_decay |
|---|---|---|---|---|---|---|
| INT8 | SAWB+ | MinMax | FP32 | – | – | – |
| INT8+50%Sp | SAWB+ | MinMax | FP32 | – | – | – |
| INT4 | SAWB+ | PACT | FP32 | max | 1e-2 | 7e-3 |
| INT4+50%Sp | SAWB+ | PACT | INT8+50%Sp | 99.9 | 1e-2 | 3e-2 |

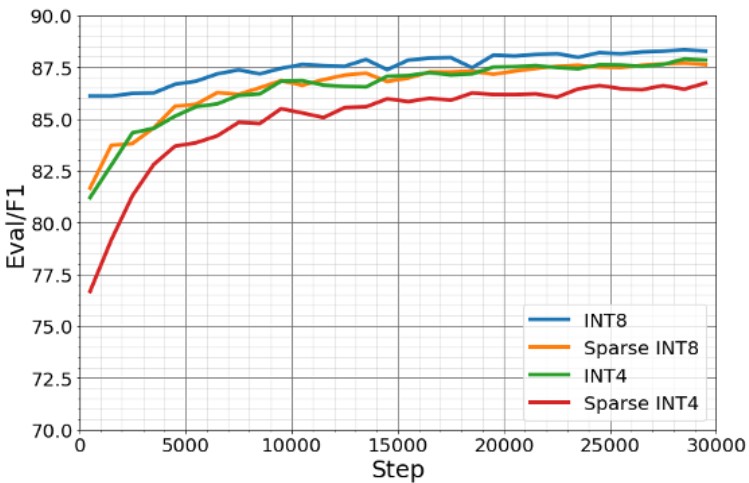

Figure 5: Convergence curves of INT8, sparse INT8, INT4 and sparse INT4 BERT-base models on SQuAD1.1.

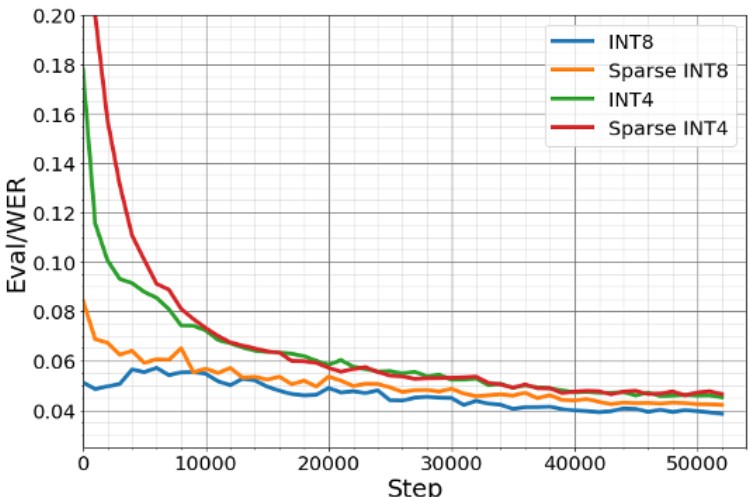

Figure 6: Convergence curves of INT8, sparse INT8, INT4 and sparse INT4 Wav2vec2.0-large models on Librispeech.

# C Appendix-C: Broader Impact

Dedicated hardware accelerators for DNN inference, including CPUs, GPUs, TPUs and other AI platforms, have powered the deployment of machine learning for real-life applications in both cloud and edge devices. Reduced precision innovations (FP16, FP8 and INT8), together with sparsity, have recently improved the capability of these accelerators by 4-8× and have dramatically improved energy cost and carbon emissions. Although pre-trained transformers have unlocked the power of transfer learning and are leading to breakthroughs in multiple application domains, the architecture is too complex for many production systems, such as those for edge-computing inference. There are many ongoing efforts to reduce the size of these models while retaining model performance and transferability. Deep compression of transformers, which is presented in this work, aims to push this front aggressively to enable faster and cheaper inference systems for a wide spectrum of deep learning models and domains. We believe that sparse 4-bit inference solutions can accelerate ML deployment and provide significant cost and energy savings for corporations and research institutes — in addition to helping reduce the carbon / climate impact of AI inference. By improving power

Table 4: Qantization/Sparsity-aware fine-tuning setting for ViT-base on ImageNet1k. Sp is short for sparsity.

| Precision Sparsity | Weight Quantizer | Activation Quantizer | Initialization Model | Percentile (%) | $\alpha\_lr$ | $\alpha\_decay$ | Epoch |
|---|---|---|---|---|---|---|---|
| INT8 | SAWB+ | MinMax | FP32 | – | – | – | 8 |
| INT8+50%Sp | SAWB+ | MinMax | FP32 | – | – | – | 8 |
| INT4 | SAWB+ | PACT | FP32 | 99.9 | 1e-2 | 1e-5 | 16 |
| INT4+50%Sp | SAWB+ | PACT | INT8+50%Sp | 99.9 | 1e-2 | 1e-6 | 16 |

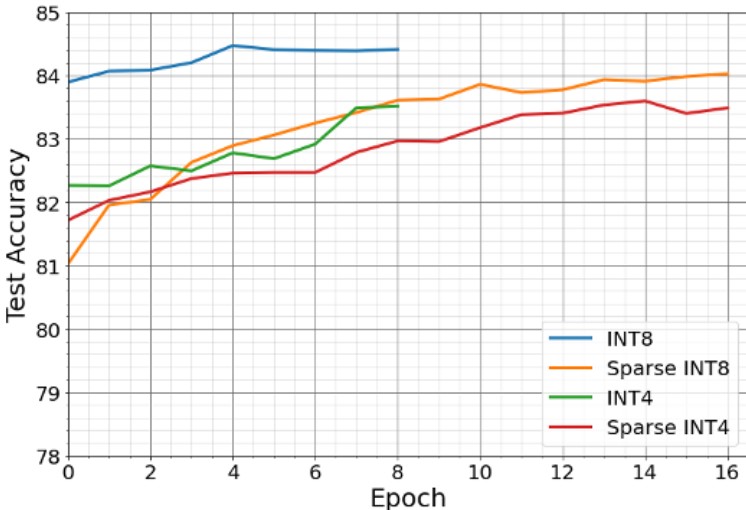

Figure 7: Convergence curves of INT8, sparse INT8, INT4 and sparse INT4 ViT-base models on ImageNet1k.

efficiency by about $4\times$ over current transformers running in FP16 (and $8\times$ vs. default FP32 designs), the carbon footprint for predicting with large DNN models can be significantly reduced [10].

The reduction in computational energy and memory footprint could also enable the inference of large transformer models to be carried out on edge devices (mobile platforms, health care devices, security cameras, consumer drones, etc.). This, in turn, could alleviate security and privacy concerns of sending data back to the Cloud for prediction tasks.

We would also like to emphasize that although we have shown promising results and limited accuracy loss in comparison to FP32 downstream tasks, deeply compressed transformer models using our solutions could still be subject to unexpected instabilities. This may necessitate a careful examination of these optimization techniques and numerical formats over a wider range of models and perfected alongside the development of ML model research. The risk of using deeply compressed transformer models in real inference applications is most likely higher than full precision dense models and thus requires task-specific robustness studies to prepare these models against adversarial attacks. In addition, More work is also needed to assess the impact of deeply compressed models in fairness and explainability.

Table 5: Uncertainty study of three sparse INT4 models. The means and standard deviations are calculated using 5 runs with different initial seeds.

| Precision Sparsity | BERT-base (F1%) | Wav2vec2.0 (WER %) | ViT (Accuracy %) |
|---|---|---|---|
| INT4+50%Sp | $87.04 \pm 0.10$ | $4.63 \pm 0.06$ | $83.65 \pm 0.12$ |

In addition, for each application domain, we chose representative models, datasets, and tasks (SQuAD1.1, Librispeech, and ImageNet1k) that have been used widely to evaluate compression algorithms. Going forward, it will be important to study different (simple and complex) downstream tasks across multiple datasets within each domain; this will be the focus of our future work. We will study various datasets for each task to assess the generalization of our proposed techniques.