# OpenReview forum: "Deep Compression of Pre-trained Transformer Models"
_NeurIPS.cc/2022/Conference — NeurIPS 2022 Accept_

### Official Review · Reviewer_CPyh · 2022-07-10

**Rating:** 7
**Confidence:** 3
**Soundness:** 3 good
**Presentation:** 4 excellent
**Contribution:** 3 good

**Summary:**

This paper presents authors' investigation on compressing pre-trained transformers with quantization/sparsity-aware fine-tuning and optimized quantizers, initialization, data format etc. Experimental results on NLP, speech and vision tasks show effectiveness of the proposed approach.

**Questions:**

1. In this paper authors list limitations with teacher-student distillation (e.g. lines 192-195). Still, teacher-student distillation could be applied to train a student model with a lower-complexity architecture, and its overhead in training/tuning may not necessarily be a blocker for using it, and it has no direct impacts on inference. Do authors have additional reasons why not consider teacher-student distillation for compression?

2. Singular value decomposition (SVD) is also widely applied for model compression, and could be easily combined with other model compression techniques. Do authors have a say on comparing/combining it with the proposed approach?

3. In check list it's said "key results are averaged results over random seeds". For this case, could error bars be included directly in results tables?

**Strengths And Weaknesses:**

The topic of compressing transformer models is of broad interests in the machine learning community, given transformer-based models are being widely researched in many areas, and it's also important to compress models for resource constraint use cases. This paper proposes an effective approach which is generally applicable to different areas. This work is technically sound, with carefully designed experiments and analysis. This paper is clearly written as well.

There are different approaches to compress models, and it would be helpful to provide more contexts on the chosen quantization + sparsity vs. other techniques (see questions below).

---

> ### Author Response · Authors · 2022-08-01
> **Response to reviewer CPyh**
>
> Thank you very much for your review and thoughtful comments. We are revising our draft and will address your concerns and suggestions.
>
> To answer your questions and comments:
>
> 1). **Distillations**: We agree distillation is a powerful tool. In fact, some of the state-of-the-art works rely on distillation to improve the performance of quantized models. We would argue that our methods are more straightforward, easier to implement, and can achieve good performance while avoiding some of the limitations of distillation such as the requirement of storing and computing the teacher models. More importantly, we do not think our methods and distillation are mutual exclusive. We believe that we can further improve the model accuracy by combining our methods with distillation, which will be our future work. We will add a section in the Related Work section to explain how our contributions complement and/or address the limitations of distillation.
>
> 2). **SVD**: SVD is another widely used compression technique that can be combined with our methods. To apply SVD on quantized and sparsified tensors can be a very interesting research direction. We thank the reviewer for the great suggestion.
>
> 3). **Error bar**: By the time of submission, we did not have enough samples for a systematic analysis on the uncertainty, so the error bar was not included in the result table. To address your question, for all three sparse INT4 models, we now perform 5 runs with different initial seeds. The means and standard deviations calculated using the 5 runs are shown in the table below. The results are consistent. However, repeating 5 runs for all the precision/sparsity settings of three models will take considerable GPU time. We will make a clarification and do a thorough uncertainty analysis in the revised version.
>
>
>
> | Precision/sparsity  |  Bert-base  (F1 %)	   |    Wav2vec2.0 (WER %) | ViT (Accuracy %)|
> | :---                         |    :----:                               |          :---:                         |    :---:                  |
> |INT4+50%sparse	|87.04±0.10	                   |       4.63±0.06	             |  83.65±0.12         |
>
>
> Table. Uncertainty study of three sparse INT4 models. The means and standard deviations are calculated using 5 runs with different initial seeds.

---

### Official Review · Reviewer_v8Zh · 2022-07-11

**Rating:** 8
**Confidence:** 2
**Soundness:** 3 good
**Presentation:** 2 fair
**Contribution:** 4 excellent

**Summary:**

The study makes a substantial contribution to more equitable and accessible natural language processing, computer vision, and speech signal processing by introducing compression methods that allow deploying the leading Transformer models up to 16 times less compute (i.e., model simplification using less bits) while maintaining model accuracy.

**Questions:**

* Remember to introduce all acronyms used in the paper at their first occurrence (in particular, Section 1 could be enhanced in this way).

* You seem to have missed a paragraph break in Section 1 before the following topic sentence: "In summary, we make the following contributions towards accurate..."

* Please elaborate in Section 3 Experimental Setup and Results on the described experiments by their further justifications. Why demonstrating impressive compression rates in three different data modalities (i.e., text of SQuAD1.1, speech of Librispeech, and images of ImageNet1k) is the way to go as opposed to, let us say, compressing on text from three different datasets (e.g., different tasks/problems/levels of difficulty) for natural language processing?

* Moreover, a more explicit addressing of the authors' right to use these datasets and methods for the purposes of this study in Section 3 Experimental Setup and Results would have been beneficial what comes to "Ethics Review Area" flagging below.

* Consider dividing this section to methods and results more clearly to ease the reader in feeling convinced that the experiments are correct and complete prior to presenting the results. I was little confused what the compression goals were and where the difference between high, excellent, outstanding, and groundbreaking were in general and in particular for each domain/task/modality addressed.

* A more traditional Discussion section before the conclusion would have probably worked better. Namely, some further revisions are needed to discuss the limitations and broader impact of this study and its contributions before concluding it.

* Double check that capitalisation is consistent in all section headings


**Limitations:**

The authors have not addressed the limitations and potential negative societal impact of their work. As stated above, a more traditional Discussion section before the conclusion would have probably worked better. Namely, some further revisions are needed to discuss the limitations and broader impact of this study and its contributions before concluding it.

**Strengths And Weaknesses:**

The paper seems to make substantial contributions and is easy to follow, although I am not an expert in compressing models. What could be clarified is the experimental design and its justification. For example, it would be beneficial for the reader to better understand why demonstrating impressive compression rates in three different data modalities (i.e., text of SQuAD1.1, speech of Librispeech, and images of ImageNet1k) is the way to go as opposed to, let us say, compressing on text from three different datasets (e.g., different tasks/problems/levels of difficulty) for natural language processing. A clearer articulation of the compression goals would have been helpful to better evaluate the achieved compression rates; I felt uncertain what high, excellent, outstanding, and groundbreaking success would look like in general and in particular for each of the three experimented modalities. That is, some more justifications are needed to truly acknowledge and appreciate experimental contributions made in this study. Moreover, a more explicit addressing of the authors' right to use these datasets and methods for the purposes of this study would have been beneficial what comes to "Ethics Review Area" flagging below. Finally, the end of the manuscript calls for some further revisions to discuss the limitations and broader impact of this study and its contributions before concluding it.

---

> ### Author Response · Authors · 2022-08-01
> **Response to reviewer v8Zh**
>
> Thank you very much for your insightful suggestions and positive comments. We are revising our draft and will address your concerns and suggestions.
>
> To answer your questions/comments:
>
> 1). **Motivation on choosing the three different data modalities**:  We agree that we should have addressed the motivation better on choosing the three different data modalities. One of the powerful features of transformer models is their versatility. Pre-trained transformer models can be applied to different application tasks (text, speech and vision), using the same attention architecture, still achieving excellent performance in each area. This brings an opportunity to greatly simplify the software and hardware-based acceleration and deployment of such deep learning models through quantization and pruning. For example, one can design and optimize a single transformer accelerator for most popular deep learning tasks. Moreover, multimodal transformer models that can simultaneously handle multiple types of data, like raw images, video and language, are recently becoming popular (reference 1). Thus, a general compression methodology for transformers that can work across different domains can be cost effective and very useful in practice. However, this goal can be challenging, since the input signals are very different which can cause a wide numerical range in the distribution of tensors across various models and modalities. To address this challenge, we introduce the methods and techniques in this paper that can help to achieve this goal.
>
> For each application domain, we chose the representative models, datasets and tasks ((i.e., SQuAD1.1, Librispeech, and ImageNet1k) which have been used widely to evaluate compression algorithms in their own area (as shown in the references in section 4, Related Work). However, as you pointed out, it is as important to study different (simple and complex) downstream tasks across multiple datasets within each domain. This will be the focus of our future work. We will study various datasets for each task to assess the generalization of our proposed techniques.
>
> 2). **Address the right to use the dataset**: We are using all public datasets and models that have been widely used to evaluate deep learning algorithms. We obtained the models from open-source repositories with proper references (Appendix-A). We will make sure to address it clearly in the revised version.
>
> 3). **Broader impact and limitation**: We do have a full section on broader impact and limitations. Due to the page limit, we put in the appendix. Please refer to Appendix-C.
>
> 4). **Format and presentation**: Thank you for all the suggestions and pointing out the errors. In the revised version, we will make the changes accordingly, including explaining the acronyms when first used, making the format and capitalization of section headings consistent, adding the missing paragraph break, and re-organizing the paper for clearer presentation.
>
> **Reference**:
>
> 1). VATT: Transformers for Multimodal Self-Supervised Learning from Raw Video, Audio and Text (<https://arxiv.org/abs/2104.11178>)

---

### Official Review · Reviewer_Q6aD · 2022-07-11

**Rating:** 7
**Confidence:** 2
**Soundness:** 3 good
**Presentation:** 3 good
**Contribution:** 3 good

**Summary:**

This paper proposes to reduce the computational requirement and memory usage of the state of the art transformer based models. A novel quantization and pruning strategy with proper initialization is proposed and number of experiments support the claim of the paper.

**Questions:**

- I think it is helpful to elaborate more on the process of fine tuning . You could maybe make a figure with some blocks visualizing the pretrained model, the customized model and explain the fine tuning and of course how the proposed method is employed.
- In general, it is more clear to report the relative improvements in the results. For example, in line 88 it is stated that the model achieved less than %1 accuracy loss; this 1% should be read next to the original accuracy.
- Regarding figure 3, it is stated that outliers play an important role in transformer training. Could we also say that the close to mean values are not as important?
- Some comments regarding the text:
  - what is q in equation 2?
  - Could you provide a reference for the zero alignment explained in 2.1.1?
  - in line 114: can be find --> can be found
  - Table 1 caption: accuracy --> performance (you are not reporting accuracy in this table)

**Strengths And Weaknesses:**

Strength:
- A novel method is presented.
- An important and challenging task is addressed

Weakness:
- The text could be improved

---

> ### Author Response · Authors · 2022-08-01
> **Response to reviewer Q6aD**
>
> Thank you very much for your thoughtful review and all suggestions to make the paper clearer. We will revise our draft to consider all comments and suggestions.
>
> To answer your questions/comments:
>
>
> 1). **Add a schematic to illustrate the fine-tuning process**: It is a great suggestion! We add a schematic and the figure caption as suggested (Fig. 4 in Appendix-B, Section 2.2). We will insert the figure in the revised version.
>
> 2). **“Less than 1% accuracy loss”**:  Since the three models/tasks use different metrics (accuracy and error rate), we did not list all baseline numbers in this summary section. We will change the line to “less than 1% accuracy loss with respect to the baseline” to make it clearer.
>
> 3). **Large outliers**: For very low precision quantization (such as INT4), a small dynamic range is preferred in order to lower the quantization error. To narrow down the dynamic range, a common method is to clip off large outliers, which usually has no impact on model accuracy and has been working well for CNN models (please refer to reference 1 below). In transformer models, however, we find that precautions are needed when clipping large outliers since they seem to play an important role in the learning process as shown in figure 3. Close-to-mean values are important as they constitute the majority of the tensor, however, a few large outliers (>99.9 percentile) that were considered much less impactful in previous works can now play an important role in transformer models. We will make the explanation clearer in the revision.
>
> 4). **q in equation2**:  Sorry, we missed that. The q is for “quantized”, Wq is the quantized weight. We will explain that clearly in the text explaining the equation.
>
> 5). **A reference on the zero-alignment**: Reference 2 below has detailed discussion on the representation of zero in quantization and its overhead in hardware implementation. We will include this reference in the revised version.
>
> 6). **Table 1 caption**: Will change to “performance” in the revision as suggested.
>
>  **References**:
>
> 1). Balanced Quantization An Effective and Efficient Approach to Quantized Neural Networks
> : (<https://arxiv.org/pdf/1706.07145.pdf>)
>
> 2). Quantizing deep convolutional networks for efficient inference: A whitepaper (<https://arxiv.org/abs/1806.08342>)

---

### Meta-Review · Area_Chair_uG2C · 2022-08-31

**Recommendation:** Accept
**Confidence:** Certain

**Metareview:**

The paper presents quantization and sparsifying (50%) for transformers, and studies simple schemes for both quite extensively. The novelty is rather low but the value of the paper is in presenting results in 3 domains (NLP, ASR, image classification) and in paying attention to details of what matters to do quantization on pre-trained models with minimal loss of accuracy, as well as efficient implementation of those schemes. Overall, the paper is suitable for NeurIPS and I recommend acceptance.

**Award:**

No

---

### Decision · Program_Chairs · 2022-09-14

Accept